# Genomics Confirm an Alarming Status of the Genetic Diversity of Belgian Red and Belgian White Red Cattle

**DOI:** 10.3390/ani11123574

**Published:** 2021-12-16

**Authors:** Roel Meyermans, Wim Gorssen, Nadine Buys, Steven Janssens

**Affiliations:** Center for Animal Breeding and Genetics, Department of Biosystems, KU Leuven, Kasteelpark Arenberg 30—Box 2472, 3001 Leuven, Belgium; roel.meyermans@kuleuven.be (R.M.); wim.gorssen@kuleuven.be (W.G.); nadine.buys@kuleuven.be (N.B.)

**Keywords:** red cattle, *Bos taurus*, genetic diversity, inbreeding, runs of homozygosity, admixture, genomics, single nucleotide polymorphism, SNP, conservation

## Abstract

**Simple Summary:**

Genetic diversity of livestock is vitally important as it enables the adaptation of future populations to changing environments. Therefore, preserving a sufficiently large genetic diversity is key. However, for many local cattle populations, little is known about their genetic diversity such as inbreeding level, effective size etc. We studied the genetic diversity of two local Belgian red cattle populations (Belgian Red and Belgian White Red cattle) using state-of-the-art genomic techniques. These tools assessed diversity at the population and individual level, and allowed the positioning of these two breeds in an international context of 52 other (European) cattle breeds. Accordingly, we contribute to the general knowledge of European red cattle, and more specifically we help the breeders, breed organization and the government to manage the genetic diversity of both breeds.

**Abstract:**

Genetic diversity is increasingly important for researchers and society. Small and local populations deserve more attention especially, as they may harbor important characteristics. Moreover, small populations are at greater risk and their genetic management is often more challenging. Likewise, European red cattle populations are threatened, as they are outcompeted by more specialized cattle breeds. In this study, we investigate the genetic diversity of two local Belgian red cattle breeds: Belgian Red and Belgian White Red cattle. A total of 270 animals were genotyped via medium density SNP arrays. Genetic diversity was assessed using runs of homozygosity screening, effective population size estimation and F_st_ analyses. Genomic inbreeding coefficients based on runs of homozygosity were estimated at 7.0% for Belgian Red and 6.1% for Belgian White Red cattle, and both populations had a low effective population size (68 and 86, respectively). PCA, F_st_ and admixture analyses revealed the relationship to 52 other international breeds, where they were closest related to some Belgian, French, Scandinavian and Dutch breeds. Moreover, F_st_ analyses revealed for Belgian Red cattle a signature of selection on BTA6, adjacent to the *KIT* gene. This study gains important knowledge on the genetic diversity of these two small local red cattle breeds, and will aid in their (genetic) management.

## 1. Introduction

Recently, we have seen a catch-up in studies of the genetic diversity of local cattle populations [1,2,3,4], a trend that is facilitated by the availability of affordable genome-wide single nucleotide polymorphism (SNP) marker panels [5,6]. On top of that, as an increasing amount of international genotype datasets become available, the strength and scope of analyzing just a few local livestock breeds improves [4,7,8]. Hence, this provides a good opportunity to study (small) local populations and situate them in an international context, providing important insights in the breed composition and potential admixture of other populations. Meanwhile, the historical architecture of breeds can be studied, whereas it also contributes to the improvement of (genetic) management of these local populations. Unfortunately, population sizes for these local breeds are often low, which hampers conservation, let alone genetic improvement.

Two local breeds that have never been genetically characterized before are the Belgian Red (BR; also called West-Flemish Red) cattle and the Belgian White Red (BWR; also called East-Flemish White Red) cattle, both from the northern part of Belgium (Flanders). These breeds categorize within the lowland red breeds of West and North Europe [9]. BR cattle are subdivided into a meat type and a dual-purpose type population (here abbreviated as BRM and BRD, respectively). BR is mostly uniform dark red, although the strict regulation for this color pattern was lifted in the 1970′s and now also red pied animals are allowed [9]. The BRM population approximately counts about 1000 registered animals and is exclusively used for beef production [10]. BR meat acquired the European Protected Designation of Origin label in 2019 (“*Vlees van het rood ras van West-Vlaanderen*”). The BRD population counts approximately 350 heads [10]. They are bred for milk production in more extensive grazing conditions, meanwhile maintaining decent beef qualities. It is assumed that (at least part) of the population is admixed with old Northern French populations (e.g., Cassel), Red Holstein and Red Danish cattle [9,11]. The BWR cattle population is approx. 1900 heads large, with a limited number of active purebred sires (<50) and is bred as a dual-purpose breed. The population has undergone some import from Red Holstein and Ayrshire [11]. Felius reports that the breed has also seen influences of Shorthorn, Durham and MRY cattle [9]. According to FAO criteria, the BRD population is recognized as “at risk: critically maintained”, whereas BRM is labelled as “at risk: endangered maintained” and BWR “At risk: endangered” [11]. To put these endangerment statuses into perspective, the European Farm Animal Biodiversity Information System estimates that 11% of European cattle populations are not at risk, while 48% are considered vulnerable to critical and 21% are already extinct (21% with unknown status) [11]. 

The other local cattle breeds in Belgium are Belgian Blue (with dual-purpose and meat-type subpopulations), East Cantons Red and White and Campine cattle (CAM). CAM cattle have already been studied after their revival in 2012 [12]. There, it was found that the breed showed a large amount of variation- with an introgression of mainly Meuse-Rhine-Yssel, Deep Red and Holstein- and only a limited number of herds seemed to harbor the original population. 

European red cattle populations are gaining more and more scientific attention. Both Schmidtmann et al., (2021) and Zinovieva et al., (2021) studied the genetic background of red cattle in Northern Europe (Germany, The Netherlands and Denmark) and Belarus, but quite a number of breeds are still out of the picture [3,4]. Red cattle are known to be well-adapted to their environment, which makes them suitable for less intensive production systems [13]. Besides, they are found to be more easy to manage and have good resilience (e.g., legs, claws, udder) [13]. However, they are also facing some challenges as they are often out-performed in intensive systems by more productive breeds (e.g., Holstein for dairy production or Belgian Blue for beef production). Therefore, their continuation is often threatened and (in situ) preservation is valuable [13]. 

The genetic diversity of BR and BWR cattle and their relation to other European cattle populations is currently unexplored. Therefore, we use state-of-the-art methods and techniques to study the genetic diversity of both the BR and BWR. In this research we use the conventional measures of genetic diversity and compare both populations to available genotyped European cattle. Hereby we complement the knowledge and genetic picture of Belgian local, red cattle.

## 2. Materials and Methods

### 2.1. Animal Sampling

Blood or semen samples from 91 BR cattle (of which 54 were BRM and 37 were BRD; seven breeders) and 179 BWR (fifteen breeders) were provided by the herdbook organization (SDVR/CRV). This set contained 27 BR artificial insemination (AI) bulls, and 31 BWR AI bulls. All cows were born between 2006 and 2018, whereas AI bulls were born between 1991 and 2017. Samples were selected by the herdbook proportionally to the farmers’ herd size in order to obtain a representative set from the active population (on average 10 animals sampled per farm/breeder). 

### 2.2. Genotyping and Quality Control

Genotyping was performed for BR on the Illumina BovineSNP50 genotyping array (53,218 SNPs) and for BWR on the EuroGenomics MD BeadChip array (41,949 SNPs). Genotype quality control was performed using *PLINK 1.9* [14]. All individuals passed quality control with >90% genotyping rate and no outlying heterozygosity (>3SD). Only SNPs with known genomic location, on autosomes and with high call rate (>95%) were withheld for the analysis. No minor allele frequency or linkage disequilibrium pruning was applied to obtain reliable estimates of ROHs as described in [15]. After quality control, genotypes from both arrays were merged resulting in 38,563 SNPs. 

### 2.3. Genetic Diversity 

Genetic diversity was assessed by first estimating N_e_ following [16,17,18], a method based on linkage disequilibrium. Average homozygosity within the studied population was calculated using *PLINK* (*--het*). A run of homozygosity (ROH) analysis was performed in *PLINK* using the scanning window algorithm (*--homozyg*). The minimal ROH length was set at 1000 kb, but was overruled by the more stringent minimal number of SNPs, which was determined following [19] (44 SNPs for BWR, 43 SNPs for BR). This minimal number of SNPs resulted in ROH of at least 1600 kb. No heterozygote SNPs and only one missing SNP were allowed, the sliding window length was set equally to the minimal number of SNPs (44 and 43 SNPs respectively for BWR and BR), with a minimal SNP density of 1SNP/200 kb, a maximal gap between two consecutive SNPs of 1000 kb and a threshold value of 0.05. The inbreeding coefficient based on ROH (F_ROH_) was estimated using the genome coverage method [15]. Besides, F_ROH > 5 Mb_ and F_ROH > 16 Mb_ were estimated with ROHs longer than 5 MB and 16 Mb, respectively. These measures give an indication of the inbreeding that occurred up to ten and up to three generations ago [20]. Genome-wide overviews of ROH incidence were visualized using the *qqman* R package [21]. ROH islands were identified according to [22]. An SNP-based Weir and Cockerham’s F_st_ analysis was performed using *PLINK (--fst; --family*) and was visualized using *qqman*.

### 2.4. Inter-Breed Analysis

To study our two Belgian cattle breeds in an international context, online available—open access SNP genotypes were used. The data were collected from [12,23] and the WIDDE database [24,25,26,27,28] and comprised of 1707 animals from 52 different populations. From the WIDDE database, only taurine (*Bos taurus*) breeds from Europe were selected. A detailed overview of all available populations is given in Appendix A, including all used abbreviations. Identical quality control measures were performed on these data, and when joined, 28,903 common SNPs remained. Principal component analyses (PCA) were performed using *PLINK* (*--pca*). Ancestry was analyzed using *ADMIXTURE* [29] and visualized using *Pophelper 2.2.7* [30]. The *ADMIXTURE* algorithm identified a group of similar animals which it considers as pure ancestors. The algorithm might give biased results when comparing populations with large discrepancies in the number of genotyped individuals. In the public dataset, the Improved Red (IR), Ringamåla cattle (RMC) and Red Holstein (HOL_R) populations were represented with the lowest number of individuals (6, 13 and 17, respectively). Therefore, a maximal number of 40 randomly selected animals per population was retained for this analysis. Finally, a neighbor joining tree and a neighbor net graph was constructed based on pairwise Weir and Cockerham’s F_st_ values (*--fst* in *PLINK*) and visualized using *SPLITSTREE5* [31]. 

## 3. Results

### 3.1. Genetic Diversity

The genetic diversity of both BWR and BR cattle was studied following the conventional genetic diversity measures. First, the N_e_ was estimated at 35 for BWR and at 21 for BR (for BRM at 14 and BRD at 11 when subdivided). After adjustment for the relative small sample sizes [17], corrected N_e_ were estimated at 86 for BWR and 68 for BR (58 for BRM and 71 for BRD). Second, ROH and F_ROHs_ were studied for both populations and are summarized in Table 1. In BR, 24.6 ROH fragments were detected on average per animal with a mean length of 6.9 Mb. In BWR, the average genotyped animal had 11.7 ROHs identified (with a mean length of 8.0 Mb each). F_ROH_ was estimated higher in BR (7.0 %) than in BWR (6.1 %), and higher in BRM (7.7 %) compared to BRD (6.1 %). The longest ROH (86.5 Mb) was detected in one BRM animal on BTA7, spanning more than 75% of the chromosome. In addition, F_ROH > 5Mb_ was highest in the BRM population, suggesting recent inbreeding up to 10 generations ago [20]. F_ROH > 16Mb_ were similar for all populations. Correlations between F_ROH_ and F_ROH > 5Mb_ were for all populations estimated between 0.97 and 0.99. Between F_ROH_ and F_ROH > 16Mb_, correlations were 0.86 for BWR, 0.79 for BRM and 0.74 for BRD. For F_ROH > 5Mb_ and F_ROH > 16Mb_, they were estimated at 0.88 for BWR and BRM, and at 0.77 for BRD. Figure 1 shows the incidence plots for SNP in a ROH for both BR and BWR cattle. In both populations, a ROH island was detected on BTA6. For BR, 41% of all animals show a ROH near the region of 65–70 Mb (in BRM for almost 50% of all genotyped animals). Table 2 puts the results for BR and BWR next to the results reported for CAM [12]. Third, F_st_ values were calculated to scan loci that genetically discern BRM from BRD. Here, Figure 2 shows a clear signal at the telomere of BTA2 (around 6 Mb).

### 3.2. Interbeed Analysis

To study BR and BWR in an international context, 52 populations (1707 animals) from international repositories were merged. First, a neighbor joining tree was constructed with all 55 populations (Figure 3, left). Based on these results, 14 populations that were more closely related to BR and BWR were selected for further analysis (Belgian Blue (BWB), Campine (CAM), Charolais (CHA), Deep Red (DR), Holstein (HOL), Red Holstein (HOL_R), Improved Red (IR), Maine Anjou (MAN), Muese-Rhine-Yssel (MRY), Normande (NOR), Norwegian Red (NRC), Ringamåla (RMC), Swedish Holstein-Friesian (SHF) and Swedish Red (SRC)). This selection was based on both historical links and breeds with the lowest F_st_ values to BR and BWR. Next, an F_st_-based neighbor net graph was constructed with the selected populations (Figure 3, right). Moreover, a PCA was performed on the selected set of data (Appendix A). These analyses reveal that BWR and BR are discernable as separate breeds. Consecutively, an *ADMIXTURE* analysis was performed on the subset of 14 populations. Figure 4 shows the results for K = 4 and K = 8, where K = 8 was deemed the optimal number based on 5-fold cross-validation. Finally, F_st_ values for SNPs were studied to identify potential genomic regions that distinguish (one of our) studied populations in the international context. In the case of BWR, no SNPs or loci show noteworthy results (results not shown). However, when comparing SNPs from BR to the other selected populations, a signal on BTA6 (60–70 Mb) found F_st_ values reaching 0.8 (Figure 5).

## 4. Discussion

For many decades, red cattle breeds in Europe are being outcompeted by other commercial populations. As a result, numbers are declining, putting genetic diversity at risk in many of these local breeds. In this study, two Belgian red cattle populations were described genetically and were situated among international populations. 

Genetic diversity of BR and BWR was assessed using the typical gauges: N_e_, ROH, F_ROH_ and F_st_ values. It was shown that genetic diversity is somewhat higher in the BWR compared to BR (N_e_ of 86 and 68, respectively). For comparison: N_e_ of CAM was estimated at 81 in 2017 (Table 2; [12]). This means that all populations score below the FAO guideline of a minimal N_e_ of 100 for sustainable population management [13,32]. As a result, both populations are definitely at risk. When comparing BRD to BRM, the BRM population clearly had a lower N_e_, which could corroborate the fact that BRM developed from the BR population since the 1980s [9]. The remaining BR evolved later into the BRD population. Schmidmann et al., (2021) and Zinovieva et al., (2021) do not estimate N_e_’s in their studied European red cattle populations, therefore comparison with N_e_ estimates of our two red cattle populations is impossible [3,4]. 

When evaluating F_ROH_, it was apparent that estimated F_ROH_ values were higher in BR than in BWR (Table 1). BR also had a higher variability of inbreeding estimates and had individuals with more extreme F_ROH_ estimates (up to 16.7%). Within BR, BRM showed the highest degree of inbreeding. Also, BWR and BRD had more similar estimated F_ROH_ values, although BRD showed more recent inbreeding (F_ROH > 5Mb_). The correlations between F_ROH_ and F_ROH > 16Mb_ implied that BR had seen more recent inbreeding compared to BWR, which was also visible in the estimated F_ROH > 16Mb_. When scanning for ROH (islands) across the genome, BR showed more loci where at least 20% of the genotyped population had a detected ROH (Figure 1). In BR, a ROH island was detected on BTA6 (65–70 Mb; present in more than 40% of the genotyped population), which was also seen in BWR but was less prevalent (up to 20% of the genotyped population). This region was also identified as a ROH island in other cattle populations: Maine Anjou, Hereford and in Normande cattle [22]. When analyzing F_st_ values between BR and the other studied populations in the inter-breed analysis, this same region was uncovered as a signature of selection (Figure 5). This region harbors the well-known *KIT* gene (BTA6: 70,166,692 bp–70,254,044 bp), a gene that is causal to several white/spotting color patterns and was also previously associated with selective sweeps in Western-European cattle breeds [33] and in Simmental bulls sampled from five European countries [34]. Therefore, we hypothesize that the BR breed could segregate a unique variant of the *KIT* locus. However, future analysis should be carried out to test this hypothesis. 

When analyzing the F_st_ values that discern SNPs between BRM and BRD cattle, a clear signal on BTA2 (around 6 Mb) appeared (Figure 2). At this position, *Mysotatin* (*MSTN*) is located (BTA2: 6.213.566–6.220.196), in which mutations are associated with increased muscle mass, e.g., causing double muscling in Belgian Blue cattle [35]. Meanwhile, a new set of BR cattle (*n* = 31) has been genotyped for the *nt821(del11) MSTN* mutation which showed that 10 animals were homozygous for the mutation, 17 were heterozygous mutated and 4 were genotyped as homozygous wildtype. This indicates that the *nt821(del11)* mutation that is primarily known from Belgian Blue also segregates in BR. 

For a thorough study of BR and BWR in an international context, availability of international genotypes is the key. In this analysis, we included all open access *Bos taurus* populations with a European link (WIDDE database + extensive online search). In this inter-breed analysis, it was shown that BR and BWR were clearly separate from UK populations (e.g., ANG and HFD), Italian breeds (e.g., SAR and PMT) and many French populations (e.g., NOR, MON and MAR) (Figure 3). However, they were closer related to some French populations like PRP and MAN. BR seemed to be closely related to BWB and MAN cattle that all descended from Durham and Shorthorn cattle. Next to some French breeds, the BR and BWR were more distantly related to other Belgian populations (BWB and CAM), Dutch populations (e.g., DR and IR) and some Scandinavian populations (e.g., SRC and SHF). Although it is assumed that BWR and BRD have had some introgression from Holstein and MRY, both were still highly separable from the included Holstein populations (HOL, HOL_R, SHF) and from MRY, with only a minor proportion of admixture (Figure 3, Figure 4 and Appendix A). The *ADMIXTURE* analysis (K = 8) uncovered a breed specific proportion of the genome in BR (colored in yellow), and identified large unique clusters for BWB, HOL, MAN, MRY and NOR (Figure 4). For the *ADMIXTURE* analysis, it has to be noted that by selecting only a subset of maximally 40 animals per breed, highly admixed individuals might be missed. Results were consistent over repeated analyses of randomly selected subsets (results not shown). If we would opt not to include all individuals of a specific breed in the *ADMIXTURE* analysis, we might bias its outcome. Indeed, we’ve noticed that an overrepresentation of one single breed may corrupt the analysis, as the algorithm may combine these animals of the same breed as a group with similar ancestors and therefore might contribute a cluster specifically to this common ancestry. As François et al. already indicated, the CAM breed did not cluster all that well in one single group (Appendix A) [12]. Likewise, the *ADMIXTURE* analysis shows that the subset of CAM cattle is highly admixed, more than BRD, BRM and BWR cattle (Figure 4). Therefore, we have to take into account that their position on both the neighbor joining tree and neighbor net graph was approximated, as the population’s average was used in these calculations.

Furthermore, the subdivision of BR into BRD and BRM was visible from the PCA (Appendix A). For some individuals, admixture was found between these subdivisions, which was also confirmed based on the available pedigree records. Moreover, based on the PCA we were able to discern BR from BWB, two popular Belgian beef breeds. Hence, beef from both breeds could be discerned, meaning it can be used to enforce the Protected Designation of Origin label for BR meat. 

## 5. Conclusions

This study reveals for the first time the genome-wide genetic diversity of both Belgian Red and Belgian White Red cattle and puts them in an international context. Both population’s genetic diversity is threatened, with effective population sizes below the FAO guidelines. Belgian Red seemed closest related to Maine Anjou and Belgian Blue cattle, whereas Belgian White Red appeared to be closer related to Improved Red, French Red Pied Lowland and Holstein cattle. Furthermore, Belgian Red cattle harbors a variant on BTA6, adjacent to *KIT*, that differentiates the breed from all other breeds included in the analysis. Therefore, this study contributes to the genomic evaluation of small local (red) cattle populations but benefits also from publicly available information. The results are key to aid governments, herdbooks and breeders in their breeding and conservation decisions.

## Figures and Tables

**Figure 1 animals-11-03574-f001:**
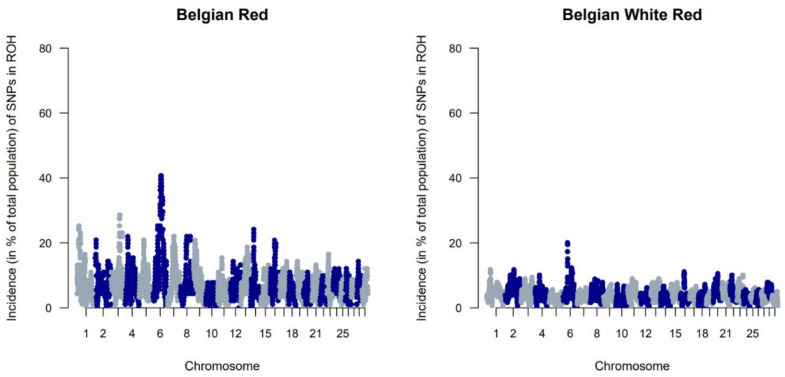
Genome-wide incidence plots of SNPs in runs of homozygosity (ROH) for both the Belgian Red (left) and Belgian White Red (right) populations.

**Figure 2 animals-11-03574-f002:**
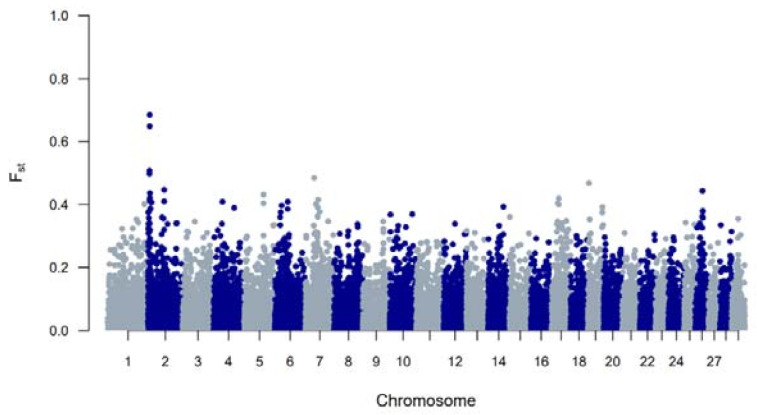
A genome-wide view of F_st_ values per SNP between the Belgian Red meat-type (BRM) and dual-purpose (BRD) populations.

**Figure 3 animals-11-03574-f003:**
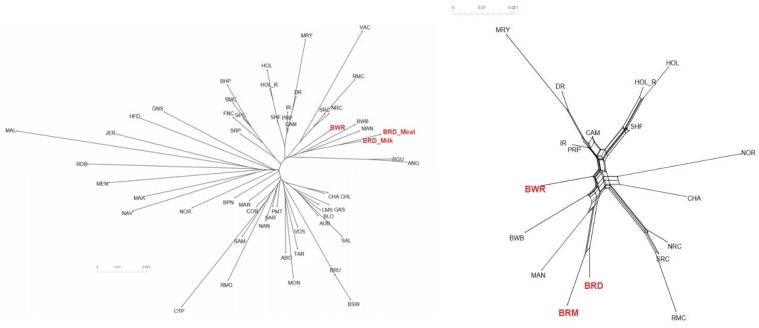
(**Left**) Neighbor joining tree of all 55 analyzed cattle populations positioning Belgian Red (meat-type: BRM; and dual-purpose: BRD) and Belgian White Red (BWR) in an international context. (**Right**) Neighbor net graph of BR (both BRM and BRD) and BWR compared to the 14 other selected breeds. Abbreviations as in Appendix A.

**Figure 4 animals-11-03574-f004:**
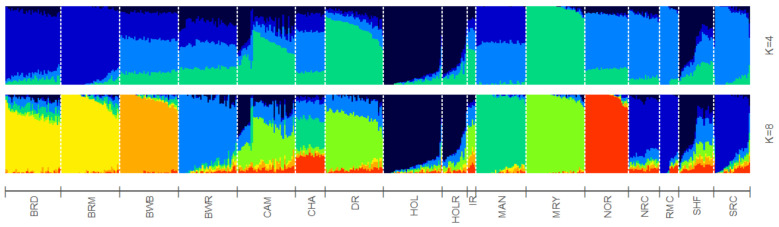
*ADMIXTRURE* clustering based on 4 and 8 clusters (K). Breed abbreviations as in Appendix A.

**Figure 5 animals-11-03574-f005:**
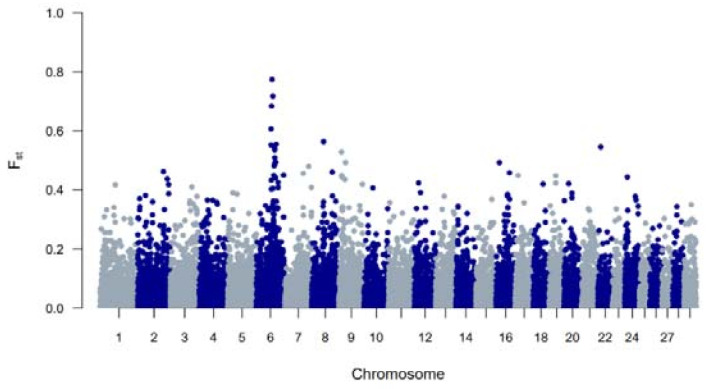
A genome-wide view of F_st_ values per SNP between the Belgian Red compared to 16 cattle populations (Belgian Blue, Belgian White Red, Maine Anjou, Charolais, Ringamåla, Norwegian Red, Swedish Red, Improved Red, Deep Red, Maas-Rhine-Ijssel, Holstein, Red Holstein, Normande, Campine and Swedish Holstein Friesian cattle).

**Table 1 animals-11-03574-t001:** Overview of the estimated ROH-based inbreeding coefficients (F_ROH_) in % for Belgian Red (BR), subdivided in the meat-type (BRM) and dual-purpose type (BRD), and Belgian White Red (BWR) cattle. F_ROH > 5Mb_ and F_ROH > 16Mb_ only takes ROH with a minimal length of 5 Mb and 16 MB, respectively, into account.

		F_ROH_			F_ROH > 5Mb_			F_ROH > 16Mb_		
		Mean	SD	Max	Mean	SD	Max	Mean	SD	Max
BR		7.0	3.1	16.7	5.3	2.8	14.4	1.9	2.0	11.5
	BRD	6.1	2.1	13.4	4.6	2.0	12.5	1.9	1.3	6.7
	BRM	7.7	3.5	16.7	5.7	3.2	14.4	1.9	2.3	11.5
BWR		6.1	2.2	13.0	3.2	2.1	11.9	1.4	1.5	9.5

**Table 2 animals-11-03574-t002:** Summary statistics of the three local red cattle breeds in Flanders (Belgium): Belgian Red (BR), subdivided in a dual-purpose type (BRD), and a meat-type (BRM); Belgian White Red (BWR) and Campine cattle (CAM). Results on the CAM breed were reported in [12] (indicated by *). Population sizes are reported by the herdbooks (year 2019–2020) [10].

	BR		BWR	CAM
	BRD	BRM		
Population size	350	1000	1900	600
Average F_ROH_	6.1%	7.7%	6.1%	4.1% *
N_e_	71	58	86	81 *

## Data Availability

The genotypes of BR and BWR cattle are deposited on the figshare repository (10.6084/m9.figshare.17025086).

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
