# Peer review of "Genomics Confirm an Alarming Status of the Genetic Diversity of Belgian Red and Belgian White Red Cattle"

_animals, 2021, doi:10.3390/ani11123574_

Round 1
Reviewer 1 Report
Manuscript ID: animals-1463112
Title: Genomics confirm an alarming status of the genetic diversity of Belgian Red and Belgian White Red cattle. Authors: Roel Meyermans, Wim Gorssen , Nadine Buys , Steven Janssens
The authors investigated in their work the genetic diversity in two local Belgian red cattle breeds using a medium density SNP arrays, thus contributing to improvement the knowledge and genetic picture of Belgian local cattle. This is an important study that can contribute to the conservation strategies of local or threatened breeds and could lead to important conclusions and a very nice publication.
Experiments have been conducted rigorously. The manuscript is presented in an intelligible fashion, technically sound, and the data support the conclusions. However, despite that the study is well performed, I am sure it can be improved.
L23-31: Please use the past tense to report what happened in the past and the present tense to express general truths, such as conclusions. This rule must be implemented for the entire manuscript
L189-190: I think that used abbreviations should be mention in text not only in Supplementary table S1
I recommend the manuscript to be considered for publication after minor correction will be performed.
Author Response
Dear reviewer,
Thank you for your review and your comment on our work. We have met all points you've suggested for improvement in our revision. You can find a point-by-point description below.
L23-31: Please use the past tense to report what happened in the past and the present tense to express general truths, such as conclusions. This rule must be implemented for the entire manuscript
We’ve checked the abstract and the whole manuscript and changed tenses where necessary.
L189-190: I think that used abbreviations should be mention in text not only in Supplementary table S1
All breeds were mentioned first before abbreviating. This was also checked throughout the whole manuscript where needed.
Reviewer 2 Report
Dear authors,
the paper does not read too well, but the topic is of interest.
Please try to be clearer in both result and discussion sections.
Line-by-line comments
Lines 17-18: you cannot help the breeds to manage their genetic diversity, you can help breeders, breeders’ associations, or governments, but not directly the breeds.
Line 31: “adjacent to KIT.” Should be “adjacent to KIT gene”.
Line 38: Please consider adding some references here about genomic tools applied to local cattle breeds (e.g., https://doi.org/10.1111/age.12697; https://doi.org/10.1186/s12863-018-0705-9)
Lines 49-50: what do you mean with “that have never been genetically characterized before”? Please look at https://doi.org/10.1111/jbg.12643
Line 65: “Felius reports that the breed also has seen influences…” should be “Felius [6] reports that the breed has also seen influences…”
Lines 88-91: These lines should be the aim of the work? Please state clearly what was the objective of your research.
Lines 96-97: “breeders” mean different “farms”? Because you have 22 “breeders” and 270 animals, but you sampled 10 animals per herd.
Lines 96-102: Were the animals connected through pedigree?
Lines 11-112: what do you mean with “genotypes from both arrays”? SNPs in common between the two?
Lines 112-113: please explain.
Lines 157-158: Table 1 summarizes just the FROH, you did not mention number of ROH per animal, ROH distribution within length class, or average ROH length in the 3 considered breeds. You are simply analyzing inbreeding coefficients based on ROH.
Lines 165-166: “SNP in an ROH” should be “SNP in a ROH”
Line 166: “an ROH island” should be “a ROH island”. Moreover, you did you decide that certain ROHs were ROH islands?
Line 167: “an ROH near” should be “a ROH near”
Line 172, Table 1: please include SD and MAX also for FROH > 5 and FROH > 16 Mb.
Line 199: Where is this “threshold of FST > 0.4” from?
Lines 223-225: this sentence has no sense. If those authors did not analyze Ne, why you should write that they did not report that??
Line 227: As aforementioned, you are not analyzing ROH, you are just analyzing ROH-based inbreeding.
Line 234: How did you chose “20% of the genotyped population has a detected ROH”?
Line 235: “clear ROH island” why clear? Someone could say that only 40% of animals had that ROH. Please declare your criterion to call the ROH islands.
Line 243: The KIT gene has been reported to be associated with white / spotting color pattern also in European Simmental bulls: some authors found this gene in a ROH island.
Line 257: “… is key” should be “… is the key”.
Line 342, Reference: please fix all the references according to the journal guidelines:
Author 1, A.B.; Author 2, C.D. Title of the article. Abbreviated Journal Name Year, Volume, page range.
Author Response
Dear Reviewer,
Thank you for your review and your insightful comments and corrections. We clarified the goal of our research, revised the description of our used methods where needed, and added the suggested results. Where necessary, we added relevant references to literature. We have spell-checked the manuscript and corrected where necessary. We’ve addressed all your comments and provide a point-by-point description below.
Lines 17-18: you cannot help the breeds to manage their genetic diversity, you can help breeders, breeders’ associations, or governments, but not directly the breeds.
We agree with this comment and have reformulated the statement.
Line 31: “adjacent to KIT.” Should be “adjacent to KIT gene”.
Done
Line 38: Please consider adding some references here about genomic tools applied to local cattle breeds (e.g., https://doi.org/10.1111/age.12697; https://doi.org/10.1186/s12863-018-0705-9)
We’ve added four recent studies using genomic tools to study (local) cattle breeds as reference to the sentence.
Lines 49-50: what do you mean with “that have never been genetically characterized before”? Please look at https://doi.org/10.1111/jbg.12643
The work of Wilmot et al. (2021) studies three local cattle breeds: the dual- purpose Belgian Blue (DPBB), the East Belgian Red and White (EBRW) and Red- Pied of Ösling (RPO). However, the EBRW cattle is different from the BWR population. EBRW cattle is officially considered extinct, and was (very) closely related to the Campine cattle in Flanders and is not (directly) related to Belgian White Red cattle (located in Flanders) (more information: e.g. https://www.regionalcattlebreeds.eu/breeds/Dual_Purpose_Red_and_White.html and François et al. 2017 Plos ONE). The existence of this population/breed is mentioned on Line 73 as “East Cantons Red”, it’s official name in FAO’s breed inventory. The Belgian White Red (also called East Flemish White Red) and the Belgian Red cattle populations were indeed never genetically characterized before.
Line 65: “Felius reports that the breed also has seen influences…” should be “Felius [6] reports that the breed has also seen influences…”
Done
Lines 88-91: These lines should be the aim of the work? Please state clearly what was the objective of your research.
We’ve clarified the goal of our research.
Lines 96-97: “breeders” mean different “farms”? Because you have 22 “breeders” and 270 animals, but you sampled 10 animals per herd.
Indeed, farms and breeders are here interchangeable. We’ve genotyped 270 animals, of which 58 were AI bulls and 212 were cows that were sampled on-farm. Thus with 22 breeders/farms we have sampled on average 10 animals per farm.
Lines 96-102: Were the animals connected through pedigree?
Yes, the pedigree is managed by the Herdbook and (limited) pedigree information was available for the sampled animals. However, this was not used during sample selection as all animals were selected as “representative” set of the breeder’s herd. Large scale pedigree data for both breeds were unfortunately not available for this research.
Lines 11-112: what do you mean with “genotypes from both arrays”? SNPs in common between the two?
Yes indeed, this was further clarified.
Lines 112-113: please explain.
This statement is omitted, as it is indeed not relevant for 2.2 Genotyping and quality control. In the genetic diversity analysis, the subdivision between meat and dual-purpose type Belgian Red cattle was made.
Lines 157-158: Table 1 summarizes just the FROH, you did not mention number of ROH per animal, ROH distribution within length class, or average ROH length in the 3 considered breeds. You are simply analyzing inbreeding coefficients based on ROH.
The average number of detected ROH per animal and the average length of these ROH are added to the discussion. However, we deem the exact number of ROH and “average ROH length” less relevant than FROH estimates, as it is possible that the ROH detection algorithm splits (wrongfully) one long ROH into two (or more) shorter ROHs. Therefore the exact number of ROH might become inflated or the average ROH length decreases, whereas the total ROH length (or % of the genome covered by ROH, thus FROH) is not affected. Therefore, we previously didn’t include these in the results section.
Lines 165-166: “SNP in an ROH” should be “SNP in a ROH”
Done, although both spellings are often used.
Line 166: “an ROH island” should be “a ROH island”. Moreover, you did you decide that certain ROHs were ROH islands?
Done. The definition of an ROH island is based on the methods given in Gorssen et al 2021 (GSE). We’ve clarified this in the Methods section.
Line 167: “an ROH near” should be “a ROH near”
Done, and checked the whole text.
Line 172, Table 1: please include SD and MAX also for FROH > 5 and FROH > 16 Mb.
We’ve added this to Table 1.
Line 199: Where is this “threshold of FST > 0.4” from?
This corresponds to the Fst value for the 0.01% upper SNPs (99.9% of all SNPs have a lower Fst value). We agree that this arbitrarily chosen limit of 0.4 has no reference in literature and therefore we have re-formulated the sentence.
Lines 223-225: this sentence has no sense. If those authors did not analyze Ne, why you should write that they did not report that??
We have reformulated this sentence. Our goal is to point out that we cannot compare Ne for these studies. Of course we don’t criticize the fact that they did not estimate Ne.
Line 227: As aforementioned, you are not analyzing ROH, you are just analyzing ROH-based inbreeding.
This is re-written.
Line 234: How did you chose “20% of the genotyped population has a detected ROH”?
As indicated above, ROH islands were identified following Gorssen et al. 2021 (GSE), which was also added in the Methods section. The 20% prevalence of the ROH island in BR and BWR is shown in Figure 1.
Line 235: “clear ROH island” why clear? Someone could say that only 40% of animals had that ROH. Please declare your criterion to call the ROH islands.
The sentence was re-written. The method for detection of ROH islands is given in the methods section as indicated above.
Line 243: The KIT gene has been reported to be associated with white / spotting color pattern also in European Simmental bulls: some authors found this gene in a ROH island.
Thank you for pointing out this interesting research. We’ve added this as reference.
Line 257: “… is key” should be “… is the key”.
Done
Line 342, Reference: please fix all the references according to the journal guidelines:
Author 1, A.B.; Author 2, C.D. Title of the article. Abbreviated Journal Name Year, Volume, page range.
This was done where necessary.
Round 2
Reviewer 2 Report
Dear authors,
I am glad that you followed my suggestions and I am satisfied by your answers and changes.
I think the manuscript is now ready to go.
Best regards